

# Toward physiological indices of emotional state driving future ebook interactivity

Jan B.F. van Erp[1,2], Maarten A. Hogervorst[1] and Ysbrand D. van der Werf[3]

[1] Perceptual and Cognitive Systems, TNO, Soesterberg, The Netherlands
[2] Human Media Interaction, University of Twente, Enschede, The Netherlands
[3] Anatomy & Neurosciences, VU University Medical Center, Amsterdam, The Netherlands

## ABSTRACT

Ebooks of the future may respond to the emotional experience of the reader. (Neuro-) physiological measures could capture a reader's emotional state and use this to enhance the reading experience by adding matching sounds or to change the storyline therewith creating a hybrid art form in between literature and gaming. We describe the theoretical foundation of the emotional and creative brain and review the neurophysiological indices that can be used to drive future ebook interactivity in a real life situation. As a case study, we report the neurophysiological measurements of a bestselling author during nine days of writing which can potentially be used later to compare them to those of the readers. In designated calibration blocks, the artist wrote emotional paragraphs for emotional (IAPS) pictures. Analyses showed that we can reliably distinguish writing blocks from resting but we found no reliable differences related to the emotional content of the writing. The study shows that measurements of EEG, heart rate (variability), skin conductance, facial expression and subjective ratings can be done over several hours a day and for several days in a row. In follow-up phases, we will measure 300 readers with a similar setup.

## INTRODUCTION

The sales of ebooks are rapidly increasing and are expected to surpass that of printed books in the near future. In its basic form, an ebook is an electronic version of the printed book. However, the devices used to access an ebook (ereader, tablet, etc.) have more capabilities than just displaying the book and turning pages on request of the reader. The device may enable true bidirectional interaction with the reader, which is a significant innovation compared to the one-directional printed book. This interactivity may substantially change the future of the ebook as artistic form and may result in new interactive media products that only slightly resemble the basic version of the printed book as sold today.

Together with scientific and cultural organizations we have started to explore the potential of interactive ebooks. One of the key questions is which reader parameter or actions (other than turning pages) are useful for interactive ebooks. One of the driving forces behind this exploration was the prominent Dutch writer Arnon Grunberg who also had a genuine interest in what his readers actually experience while reading his work, or

Corresponding author
Jan B.F. van Erp, jan.vanerp@tno.nl

more generally stated: "*Is reading a novel good for you?*" (the writer himself takes a devil's advocate stance and postulates the possibility that reading literature has a detrimental influence (*Grunberg, 2013*)). From neuroscientific data, we know that reading is a complex task involving many brain areas (*He et al. (2013)*, see *Carreiras et al. (2014)* for a recent review and *Nijhof & Willems, (2015)* for individual differences in narrative comprehension) and that reading can (at least temporarily) alter connectivity in an individual's brain (*Dehaene et al., 2015*; *Berns et al., 2013*). However, just reading text doesn't make one more social or empathetic. This may only happen after so-called "emotional transportation" (*Bal & Veltkamp, 2013*; *Johnson, 2012*), i.e., as a reader one needs to be involved at an emotional level. It is postulated that there are no effects of reading non-fiction and also no effects of reading fiction when there is no emotional transportation (*Kidd & Castano, 2013*). A similar concept (immersion) is used in the "fiction feeling hypothesis" (*Hsu, Conrad & Jacobs, 2014*) which postulates that negative, high arousal text activates the affective empathic network (*Walter, 2012*) which facilitates immersion in the text. In an experiment, participants read neutral and fearful sections of the Harry Potter saga and the results indeed showed a relation between neuronal activation pattern and subjectively rated immersion. Emotional experience is not only an essential catalyst, but also important in choosing which book to read, experiencing the content (*Johansen, 2010*) and interpreting the narrative (*Mar et al., 2011*).

All the above led us to develop a research project to measure readers' emotions while reading an ebook. Emotional state can be a key parameter to drive interactivity in future ebooks and may be viable in a real-life situation using recent Brain–Computer Interface (BCI) technology. In addition, we were interested in measuring the emotions of the writer during the writing process to be able to compare the reader's emotional state while reading a certain paragraph to that of the author during writing that same paragraph. Capturing the emotional state of the writer (both through neurophysiology and subjective ratings) became our case study and is reported in this paper to illustrate the use of sensor technology and to investigate whether prolonged physiological measurements are feasible in a real life situation. The framework described here is the basis for follow-up studies in which several hundreds of readers will read the book before publication while being measured with a similar setup as used here with the author (*Brouwer et al., 2015*). The applied, real life perspective guided the selection of theoretical models and measurement methods.

## Art, beauty and neuroscience

There is a growing interest in using neurophysiological measures to assess media, including paintings, music and films. Research in this area is still at the forefront of cognitive neuroscience and results and theoretical foundations are still under debate. An important question that has fascinated and divided researchers from both the neurosciences and the humanities is whether brain activity can provide insight in what true art and beauty is. From an applied point of view, the relevant question is whether an individual's brain pattern is informative of his or her appraisal of the piece of art. Research of Zeki and colleagues, amongst others, has shown that there is a functional specialization for perceptual versus aesthetic judgments in the brain (*Ishizu & Zeki, 2013*) and that there is a difference in

activation pattern for paintings experienced as beautiful by an individual and those experienced as ugly. This finding is independent of the kind of painting: portrait, landscape, still life, or abstract (*Kawabata & Zeki, 2004*). Hasson and colleagues (*2008*) used fMRI to assess the effects of different styles of filmmaking on brain patterns and suggest that neurophysiological sensing techniques can be used by the film industry to better assess its products. The latter was done by *Fleureau, Guillotel & Orlac (2013)* who measured skin conductance as an affective benchmark for movies and by *Golland, Keissar & Levit-Binnun (2015)* who measured cardiovascular and electrodermal signals and found a high degree of simultaneity between viewers, but also large individual differences with respect to effect size. So far, interactivity based on viewers emotional state has not moved beyond a few artistic experiments: "unsound" by Filmtrip and SARC (http://www.filmtrip.tv/) and "Many Worlds" by Alexis Kirke (http://www.alexiskirke.com/).

In this paper we look at the (applied) neuroscience behind both the creative and the emotional brain and how emotional state can be captured using wearable, mobile technology that is usable while reading an ebook. We will also explore the possibilities opened up after capturing a reader's emotional state and what the ebook of the future might look like. The paper also presents the data of the writer during the creation of emotional text (*Van der Werf & Van Erp, 2014*).

## THE EMOTIONAL BRAIN

Stimuli evoking emotions are processed in the brain through specific pathways and with the involvement of several brain areas. In other words, the emotional brain is a network of collaborating brain areas and not a single location (*Dalgleish, 2004*; *Tovote, Fadok & Luthi, 2015*). The majority of the sensory information entering the brain goes to the primary sensory areas, but a small part of the information goes to the amygdala, part of the limbic system deep inside the human brain. A main driver of the amygdala is danger: in case of a potential threat to the organism, the amygdala is able to respond quickly and prepare the body for action without much stimulus processing. The amygdala enables the release of stress hormones leading to peripheral effects, for instance increased heart rate to pump more blood to the lungs and muscles. After the amygdala, processing continues through the cingulate cortex, the ventromedial prefrontal cortex and finally the dorsolateral prefrontal cortex. Only in the dorsolateral prefrontal cortex is the processing stream through the amygdala integrated with the more cognitive processing stream from the sensory cortices. The emotional experience is a result of the interpretation of both processing routes taking into account the context and previous experiences. This integration and interpretation of information is a typical function of the prefrontal cortex (*Isotani et al., 2002*).

### Psychological framework of emotions

Before we can discuss how we can measure emotional state, we should first look into the frameworks to classify emotions. There are many psychological frameworks available. Classic work by *Ekman (1992)* and *Russell (1980)* shows that there are several basic emotions: fear, disgust, anger, happiness, sadness and surprise. This set of six basic emotions has been expanded through the years with numerous subclasses. From a neuroscientific

point of view, an important question is whether these emotions each have their own (unique) neuronal location or circuit (i.e., a discrete model (*Barrett & Russell, 1999*)), or vary along several independent dimensions (i.e., a dimensional model (*Mauss & Robinson, 2009*)), a matter that is still under debate. As described above, experiencing an emotion is the result of the integration and interpretation of numerous information streams by an extended network of brain areas which makes a discrete model unlikely. Therefore, we adopt a dimensional model, or more specifically the circumplex model of emotion (*Russell, 1980*; *Russell & Barrett, 1999*; *Posner, Russell & Peterson, 2005*) in which emotions are plotted in two dimensions: arousal and valence. For instance, anger is linked to negative valence and high arousal, sadness to negative valence and low arousal, and happiness to positive valence and high arousal and contentment to positive valence and low arousal. This model is commonly used to investigate for instance emotional words, facial expressions, and affective states.

The circumplex model of emotion stems from the ratings of individual, written words. Neuro imaging studies confirm the two-dimensional model of valence and arousal and although there may be complex interactions between both dimensions, they both have a different signature of brain activation, spatially as well as temporally. Arousing words show a different pattern (compared to neutral words) mainly in the early processing stages (i.e., within 400 ms after presentation including the following ERP components: early posterior negativity (EPN), P1, N1, P2, and N400) while the difference between positively versus negatively valenced words shows in later processing stages (between 500 and 800 ms after presentation including the late positive complex (LPC)) (*Rellecke et al., 2011*). In the spatial domain, arousal is linked to amygdala activity and valence to the cingulate cortex and the orbitofrontal cortex (*Colibazzi et al., 2010*; *Herbert et al., 2009*; *Kuchinke et al., 2005*; *Lewis et al., 2007*; *Posner et al., 2009*). Excellent reviews are given by *Kissler, Assadollahi & Herbert (2006)* and *Citron (2012)*. Based on her review, Citron (*Citron, 2012*) comes to the conclusion that positive and negative valence may differ with respect to the cognitive functions they activate and are not necessarily a continuous dimension. Although a novel is more than a collection of individual words, there has been very little research on the physiological reactions to reading larger pieces of text (see the first section of this paper), but a lot to reading individual words. This project aims to fill that void.

## Emotion classification using neurophysiological measures

With the circumplex model as point of departure, we can start to identify physiological signals that reflect the arousal and valence of emotions and that can potentially be measured while reading outside a laboratory environment. We will look at a broader range of methods used to induce an emotional state than written words and at a broader set of physiological measures than EEG and fMRI. For example, *Min, Chung & Min, (2005)* induced emotional state by letting subjects imagine pleasant, unpleasant, aroused and relaxed situations and measured effects on EEG, heartrate, skin conductance, skin temperature and respiration.

### *Valence*

Valence requires central nervous system indices as it is less clearly reflected in peripheral measures. Wearable sensors like EEG are not able to measure activity in deeper structures

like the limbic system but as reviews show (*Kissler, Assadollahi & Herbert, 2006*; *Citron, 2012*), valence is strongly linked to later processing stages involving more superficial brain structures related to cognitive processing.

Valence is reflected in (late) ERP components (*Bayer, Sommer & Schacht, 2010*; *Herbert, Junghofer & Kissler, 2008*; *Holt, Lynn & Kuperberg, 2009*), in the power in specific EEG frequency bands like alpha (*Bahramisharif et al., 2010*; *Klimesch, Sauseng & Hanslmayr, 2007*), in the relative power in different EEG bands (*Ko, Yang & Sim, 2009*) and in asymmetrical alpha activity in the prefrontal cortex (*Isotani et al., 2002*; *Fox, 1991*; *Schmidt & Trainor, 2001*; *Tomarken et al., 1992*) indicating increased left prefrontal cortex activity for positive valence and increased right prefrontal cortex for negative valence. However, power in the different frequency bands and hemispheric asymmetry are under the influence of many factors, which may only partially correspond to emotional valence. For example, hemispheric asymmetry has been linked to stress (*Lewis, Weekes & Wang, 2007*) and the tendency to approach versus to avoid stimuli (*Verona, Sadeh & Curtin, 2009*), and low power in the alpha band may be caused by the fact that stimuli with high valence may attract more attention (*Brouwer et al., 2009*; *Muehl et al., 2011*).

### Arousal

Arousal is less clearly linked to brain activation patterns except for activity in the amygdala and the reticular formation (*Posner, Russell & Peterson, 2005*), which are difficult to measure with wearable sensors like EEG. However, arousal is reasonably clearly reflected through a relatively strong activation of the sympathetic as compared to the parasympathetic autonomous nervous system. Arousal can be measured peripherally through, for instance, skin conductance (increasing conductance with increasing arousal (*Roth, 1983*)), heart rate variability (HRV), especially high frequency HRV as this is exclusively affected by the parasympathic system (reduced high frequency HRV with increased arousal (*Berntson et al., 1997*)), pupil size, heart rate (HR) and respiration frequency (all increased with increased arousal, although this pattern is not consistent over studies, see *Kreibig (2010)* for an elaborate overview).

## Current state of the art in (applied) emotion capture

The state-of-the-art in emotion detection using neurophysiological indices is that we are able to distinguish several valence and arousal levels in a lab environment when subjects are sitting still and sufficient control data is gathered beforehand to train classification algorithms (see *Van Erp, Brouwer & Zander (2015)* for an overview). However, it is important to note that the relation between physiology and emotion is not straightforward. Different studies with different stimuli and contexts report different types of correlations (*Kreibig, 2010*; *Dockray & Steptoe, 2010*). It is thus important to study relations between (neuro-) physiology and emotion within the context and under the circumstances of interest (*Brouwer et al., 2015*; *Van Erp, Lotte & Tangermann, 2012*).

An important step in this project, is to bring neurophysiological signals out of the lab and explore their potential value in daily life (*Van Erp, Brouwer & Zander, 2015*; *Van Erp, Lotte & Tangermann, 2012*). Monitoring and using the (neuro-) physiological signals of

readers is new, and entertainment in general is a good first case to transfer the technology from the laboratory to real life. This transition will come with several challenges ranging from coping with external noise due to movement artifacts, multitasking users, and usability aspects such as prolonged usage (*Van Erp, Lotte & Tangermann, 2012*; *Van Erp et al., 2013*; *Van Erp et al., 2011*). First steps in this transition have recently been made in studies investigating EEG signals in gaming (*Reuderink, Mühl & Poel, 2013*) and into music perception in realistically moving participants (*Leslie, Ojeda & Makeig, 2014*). Here we also present the case of the writer wearing physiological sensors for several hours a day and for 9 days in a row.

## THE CREATIVE BRAIN

The current case study focused on the writer and his emotional signals during the creative writing process. Our primary goal was to implement and learn about the transition from laboratory to real life before upscaling the set-up to hundreds of readers, and to capture the emotional signals of the writer as function of the emotional content of the written paragraphs. We deemed it worthwhile, nevertheless, to have a quick look at the creative brain as well. Most people would agree that creative abilities make us unique in the animal kingdom. Interestingly, we understand little of the processes that drive or facilitate creativity and still debate on the definition of creativity, although most agree upon the importance of both *novelty* and *usefulness* (see *Piffer (2012)* for an elaborate discussion).

Similar to the neuroscience of art and beauty, neuroscientific research into creativity can still be characterised as embryonic and neuroscientific models are not widely established yet. Like emotion, creativity is not related to a single brain area but rather to networks of brain areas. Based on an extensive review, *Dietrich & Kanso (2010)* even stated that "*creativity is everywhere*"; see also *Arden et al. (2010)*. Having said that, recent neuroimaging studies seem to show that creativity involves common cognitive and emotional brain networks also active in everyday tasks, especially those involved in combining and integrating information. For the current project, it is useful to distinguish two different types of creative processes as described by *Dietrich (2004)*. The first can be called *controlled creativity* often in relation to finding creative solutions for a particular, given problem. This creative process is controlled through the prefrontal cortex (*Ellamil et al., 2012*) that guides the search for information and the combination of information within a given solution space. A powerful mechanism which is bound, though, by limitations of the prefrontal cortex; for instance, with respect to the number of solutions that can be processed in working memory. The second type can be named *spontaneous creativity*, often in relation to artistic expression. This form of creativity comes without the restrictive control from the prefrontal cortex, and the process differs from controlled creativity qualitatively (e.g., solutions are not bound by rational rules like the rules of physics) and quantitatively (the number of solutions is not restricted by for instance the limited capacity of working memory). Spontaneous creativity is linked to unconscious processes (of which dreaming may be an extreme form). However, the prefrontal cortex becomes involved in spontaneous creativity when solutions will eventually reach the conscious mind, and the prefrontal cortex is required to evaluate them and bring them to further maturity.

Recent data show us that less activity in the dorsolateral prefrontal cortex links to increased spontaneous creativity in, for instance, musicians (*Limb & Braun, 2008*; *Liu et al., 2012*), and increased activation to increased controlled creativity. Results also show that there is a burst of wide-spread gamma activity about 300 ms before the moment of insight in spontaneous creativity. Gamma activity is, amongst other features, linked to binding pieces of information. A burst of gamma activity is indicative of finding (and binding) a new combination of chunks of (existing) information. *Fink & Benedek (2014)* underline the importance of internally oriented attention during creative ideation in a more general sense, reflected in an increase in alpha power.

Creativity is also linked to hemispheric asymmetry. A meta-analysis (*Mihov, Denzler & Förster, 2010*) showed that the right hemisphere has a larger role in creative processes than the left hemisphere. This is confirmed by patient research (*Mayseless, Aharon-Peretz & Shamay-Tsoory, 2014*; *Shamay-Tsoory et al., 2011*). A lesion in the right medial prefrontal cortex hinders the creation of original solutions while a lesion in the left medial prefrontal cortex seems to be beneficial for spontaneous creativity. However, experiments with creative students (*Carlsson, Wendt & Risberg, 2000*) and extremely creative professionals from science and arts (*Chávez-Eakle et al., 2007*) both show bilateral cerebellum involvement, seemingly confirming the statement that "*creativity is everywhere in the brain*."

However, these findings are general findings and may not be applicable to the creative writing process (*Shah et al., 2013*). For instance, creative writing seems to result in increased activity in the left prefrontal cortex (presumably because of its links to important language areas in the left hemisphere) except when writing emotional text, for which activity in the right hemisphere seems to be greater. This shows that the body of knowledge on the creative brain is growing but still limited and identifying neural correlates of the creative writing process requires further research. Another interesting debate is whether creative writing is a skill one can develop like skilled behavior in sports and music, or possibly even non-creative, non-fiction writing like scientists and journalists do. Lotze and colleagues found that the caudate nucleus (involved in skilled behavior) was active in experienced creative writers but not in novices (*Erhard et al., 2014*; *Lotze et al., 2014*), indicating that creative writing can indeed be a (trainable) skill.

## THE CASE STUDY

### Methods

#### *Participant*

Arnon Grunberg (http://www.arnongrunberg.com/) participated in the study. Arnon Grunberg was born in 1971 and has lived in New York since 1995. He writes novels, short stories, columns, essays and plays. His work was awarded with several national and international prizes and translated into 30 languages. He participated voluntarily, being aware that his participation would not be anonymous. All data were collected in November 2014 in Arnon's apartment in New York. The Institutional Review Board of TNO Human Factors (TCPE Soesterberg, The Netherlands) approved the study after inclusion of specific sections in the informed consent regarding privacy and data dissemination. Arnon read and signed the informed consent before data gathering began.

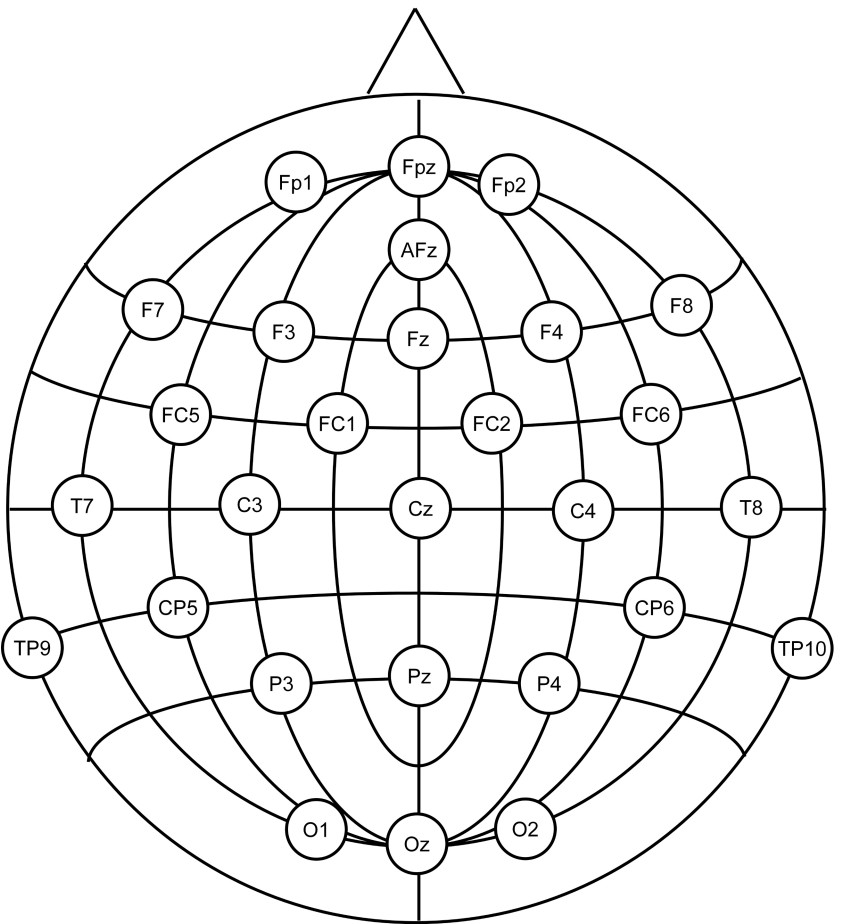

**Figure 1** Location of the 28 electrodes in the 10–20 system.

### Apparatus

We used commercially available hardware and software to record physiological parameters, facial expression and text entry. In addition, paper and pencil were used for subjective questionnaires described in the next section.

All neurophysiological signals were recorded using a wearable Mobita® 32-channel physiologic signal amplifier system sampling at 1,000 Hz (TMSi, Hengelo, The Netherlands, http://www.tmsi.com/). The available channels were used for EEG (28 TMSi water based electrodes, see Fig. 1 for the layout of the electrodes), ECG (two pre-gelled disposable TMSi snap electrodes) and Endosomatic Skin Potential ESK (pair of TMSi finger electrodes). The Mobita® has built-in accelerometers which we used to log possible activity of the writer and to synchronize the physiological data to other data gathered through a Noldus Observer XT® system (Noldus IT, Wageningen, The Netherlands; http://www.noldus.com/). This system recorded the images from two IP cameras (one providing an overview of the work space and one providing a close-up of the writer's face for later analysis of his facial expression), a continuous screen dump of the writer's PC screen, and the writer's keystrokes (Noldus uLog tool®). The writer used his normal work space and own PC, see Fig. 2.

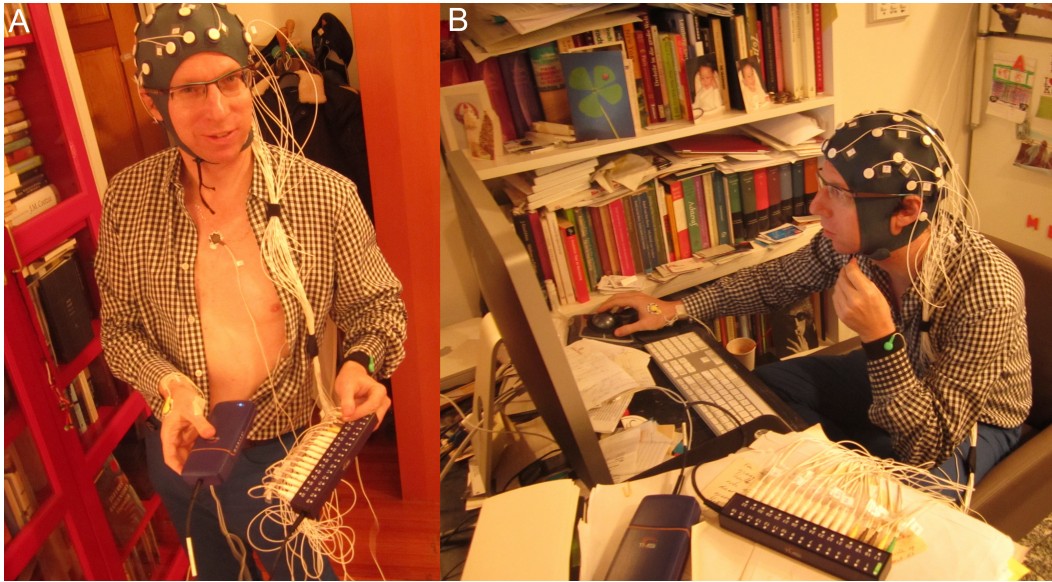

**Figure 2** Writer showing the neurophysiological sensors (A) and during writing (B).

**Table 1** Experimental protocol for one day.

| | |
|---|---|
| Start of the day | Instrumentation of the participant and check of measurement systems |
| Calibration of emotional state | Subjective questionnaires: feelings grid, VAS, DES full |
| Block 1 | Continuous monitoring of physiology and facial expression<br>Continuous logging of text entry<br>Continuous observation by experimenter<br>Subjective questionnaires at significant events[a]: feelings grid, DES self<br>Subjective questionnaires at end of block: feelings grid, VAS, DES self, DES book |
| Block 2 | Similar to Block 1 |
| End of the day | Subjective questionnaires: feelings grid, VAS, DES full<br>Open Questions |

**Notes.**

[a] Significant events could include a writer's block, or moment of great insight etc. to be identified by the participant himself. Eventually, no 'significant events' were indicated.

### Experimental protocol

Table 1 gives the outline of the experimental protocol for one day. Table 2 gives the details on the experimental protocol.

### Data processing

Our intention was to use the calibration sessions of each day to identify differences in physiological markers that could be linked to the emotional content of the written paragraph. When we can reliably establish this 'ground truth,' it could consecutively be used to analyze the data gathered during the writing blocks. After checking the synchronization between the different data streams, the physiological data of the calibration session were

**Table 2   Specification of the experimental protocol.**

| | |
|---|---|
| Instrumentation of the participant and check of measurement systems | The physiological sensors were attached to the writer and signals checked for their integrity. Video cameras, screen and keyboard logging were switched on and checked. The Mobita recorder was time linked to the Observer XT system by using the Mobita to type a specific series of key strokes recorded by the accelerometers in the Mobita, the uLog module of the Observer XT and the overview camera. |
| Calibration of emotional state | ● At the beginning of each day the following calibration data were recorded in fixed order: |
| | ○ 1 min of rest with eyes open |
| | ○ 1 min of rest with eyes closed |
| | ○ 6 blocks of 2 min filled with writing a paragraph with the following instruction: "write a paragraph on this picture with emotion ×, as if you are writing a paragraph in your novel." This instruction was accompanied with an A4 sized, full color picture from the IAPS database (*Lang, Bradley & Cuthbert, 2008*) matching emotion × (disgust, fear, sadness, amusement, contentment, excitement). We selected 60 pictures from the IAPS database, 10 for each emotion: disgust (3061, 7360, 7361, 8230, 9290, 9330, 9373, 9390, 9490, 9830), fear (1052, 1110, 1113, 1301, 1302, 1321, 1931, 3280, 5970, 5972), sadness (2130, 2271, 2312, 2490, 8010, 9120, 9190, 9210, 9331, 9912), amusement (1340, 1810, 1811, 1920, 2092, 2344, 2352, 2791, 7195, 8600), contentment (1500, 2150, 2160, 2058, 2304, 2530, 2550, 2560, 4700, 5201), and excitement (8030, 8031, 8034, 8116, 8117, 8200, 8220, 8260, 8370, 8440). The order of the emotions was balanced over the days, each picture was only used once during the experiment. |
| | ○ 1 min of rest with eyes open |
| | ○ 1 min of rest with eyes closed |
| Feelings grid | ● The feelings grid (*Russell, Weiss & Mendelsohn, 1989*) was a pen and paper A4-sized form with the instruction "Please indicate how you feel RIGHT AT THIS MOMENT. Place an "X" in the box closest to how you are feeling at this time." The form consisted of a 10 × 10 square grid with the following markers: |
| | ○ middle-top: arousal, middle-bottom: sedation, sleepiness |
| | ○ left-middle: unpleasant, right-middle: pleasant |
| | ○ left-top: anger, stress; right-top: joy excitement |
| | ○ left-bottom: depression, sadness; right bottom: relaxation, contentment |
| VAS (visual analog scale) | The VAS was a pen and paper test with the instruction "Please mark how you feel RIGHT AT THIS MOMENT." Four scales were printed on one A4: relaxed–agitated, happy–sad, optimistic–pessimistic, state of flow–no flow. |
| DES full | The DES (*Fredrickson et al., 2003*) full was a paper and pen test with the instruction "Please indicate how each emotion reflects how you feel RIGHT AT THIS MOMENT." It depicted the following 20 items on an A4-sized paper with five tick boxes to their right representing not at all (1)—completely (5): amusement, awe, contentment, gratitude, hope, love, pride, sexual desire, joy, interest, anger, sad, scared, disgust, contemptuous, embarrassed, repentant, ashamed, surprised, sympathetic. |
| DES self | The DES self was a paper and pen test with the instruction "Please mark the emotions that best reflect your feelings over the past measurement period." It contained the same 20 items as the DES full on an A4-sized form but with only one tick box to their right. |
| DES book | The DES book was the same as the DES self, except for the instruction: "Please mark the emotions that best describe the section you wrote in the past measurement period." |
| Open questions | During the debriefing, the writer answered several open questions about the use of substances (coffee, tea, cigarettes, medication etc.), the experience of flow, satisfaction about the progress, significant moments during the writing etc. The experimenter could expand the open questions based on observations made during the day. |

separated in 10 epochs corresponding to 1 min rest eyes open, 1 min rest eyes closed, 6×2 min 'emotional writing' (each corresponding to one of six different emotional pictures and descriptors), and again 1 min rest eyes open, 1 min rest eyes closed.

*EEG.*   The EEG data were processed using the following pipeline: re-referencing to channel TP10, rejection of channels with very large variance (channels O1, Oz and O2 were very

noisy and removed completely from the dataset), band pass filtering 0.5–43 Hz, and down sampling to 250 Hz. Initialy, the EEG data of the remaining channels were used in an Independent Component Analysis (ICA) to identify and remove potential artifacts. However, the ICA revealed that potential artifacts were non-stationary (i.e., changing over time) and therefore difficult to identify and thus no more data were removed. The power in different frequency bands: delta (0–4 Hz), theta (4–8 Hz), alpha (8–13 Hz), SMR (13–16 Hz), Beta (16–30 Hz) and gamma (30–80 Hz) were used as features in the classification.

*Peripheral physiology.*  As a measure of heart rate, we determined the mean interval between successive R-peaks in the ECG (RRI) for each epoch and converted this to mean Heart Rate (meanHR = 1/meanRRI). Four measures of heart rate variability were derived. The root mean squared successive difference between the RRIs (rmssdRRI) reflects high frequency heart rate variability. We also determined heartrate variability in the low, medium and high band using a spectral analysis (HRVlow, HRVmed, HRVhigh). High-frequency heart rate variability was computed as the power in the high frequency range (0.15–0.5 Hz) of the RRI over time using Welch's method applied after spline interpolation; similarly for mid-frequency (0.07–0.15 Hz) and low-frequency (0–0.07 Hz) heart rate variability. No anomalies were present in the ECG data so no data was removed. From the ESK, the mean ESK over the epochs was calculated. For the ESK we removed one outlier (contentment epoch on day 2).

*Classification analysis using EEG and peripheral physiology features.*  To determine how well various feature sets could predict the emotional state of the author during the calibration session we performed a classification analysis. Classification was performed using the Donders Machine Learning Toolbox (*Van Gerven et al., 2013*). Two types of classifiers were used: a linear Support Vector Machine (SVM) and an elastic net model with logistic regression (*Friedman, Hastie & Tibshirani, 2010*). As input we used the features that were standardized to have mean 0 and standard deviation 1 on the basis of data from the training set. One-tailed binomial tests were used to determine whether classification accuracy was significantly higher than chance.

*Facial expression.*  The images from the close-up camera were analysed offline using Noldus FaceReader software. Output for each epoch are intensity values for the following classifications: Neutral, Happy, Sad, Angry, Surprised, Scared, Disgusted.

*Subjective questionnaires.*  The data of the feelings grid, VAS, and DES full questionnaires was not pre-processed but directly analysed. We only statistically analysed the main effects of day (9 levels) and session (start of day and end of day for DES full, and start of day, end of block 1, end of block 2, end of day for feelings grid and VAS). The DES full scores were analysed using non-parametric statistics with alpha level Bonferroni adjusted for the number of comparisons. Feelings grid and VAS scores were analysed with a parametric ANOVA.

### Procedures

We started the measurements on the day the writer started with a new novella to be used in phase 2 of the project. We adjusted the measurements to his usual daily writing schedule comprising two blocks: one in the morning and one in the late afternoon or early evening. He normally writes for about two hours and fills the time in between with other activities (including other writing activities). During a writing block, he was engaged in other activities as well like answering emails and phone calls etc., but never during the instrumentation and calibration. All activities during the measurement blocks were logged by the experimenter who was always present during the measurements. We measured for nine consecutive days. At the end of the day, the experimenter and the writer would make a specific schedule for the next day. The writer also reflected on his experiences over the day, including the user experience of wearing the equipment and being observed. The day before the start of the experiment, the protocol, instructions etc. were explained in great detail, the writer signed the informed consent, his workplace was instrumented and the equipment tested. Besides the addition of the equipment, the writer's workplace was not altered in any way to give the writer the best opportunity to behave as usual. On each measurement day, the experimenter came to the apartment as scheduled and followed the protocol as detailed above. At the end of the day, all data were encrypted and saved to an external hard disk.

## Results

### Classification of baseline vs. emotion conditions in the calibration blocks

First, we determined whether the feature set contained information to discriminate the baseline conditions (Eyes Open and Eyes Closed) from the emotional (writing) epochs using binary classification (baseline vs. non-baseline). For this purpose we performed a 'leave-one-day-out' cross validation using the SVM classifier. This method is to be preferred over *random* N-fold cross validation since it better accounts for possible correlations between data during the day (*Lemm et al., 2011*). Still, the results when using random folds were found to be comparable to the results of the analysis presented here. It is also important to compensate for the imbalance in the number of conditions, with 36 baseline blocks and 54 emotional blocks in the set. All reported performance scores follow a binomial distribution and the variability of the binomial distribution follows directly from the average score and the number of measurements (the distribution is not well approximated with a Gaussian distribution and therefore the variance is not a good indicator of the variability in the results). For a larger number of measurements the variability is approximately equal to $p*(1-p)/N$ (with $p$ estimated by the score, $N$ the number of measurements).

When all six physiology features (i.e., meanHR, HRVlow, HRVmed, HRVhigh, rmssdRRI, meanESK were used as input to the classifier the average model performance (over all days)) was 71%, with a hit-rate (score for correctly classifying baseline blocks) of 58% and a False-Alarm-rate (FA-rate, i.e., fraction of falsely classified emotional blocks) of 20%, resulting in an equal cases (in the situation in which both conditions occur equally frequent) performance of 69% ($p < .01$). Individual ANOVAs with condition as independent variable (baseline vs. writing) and physiological measure as dependent

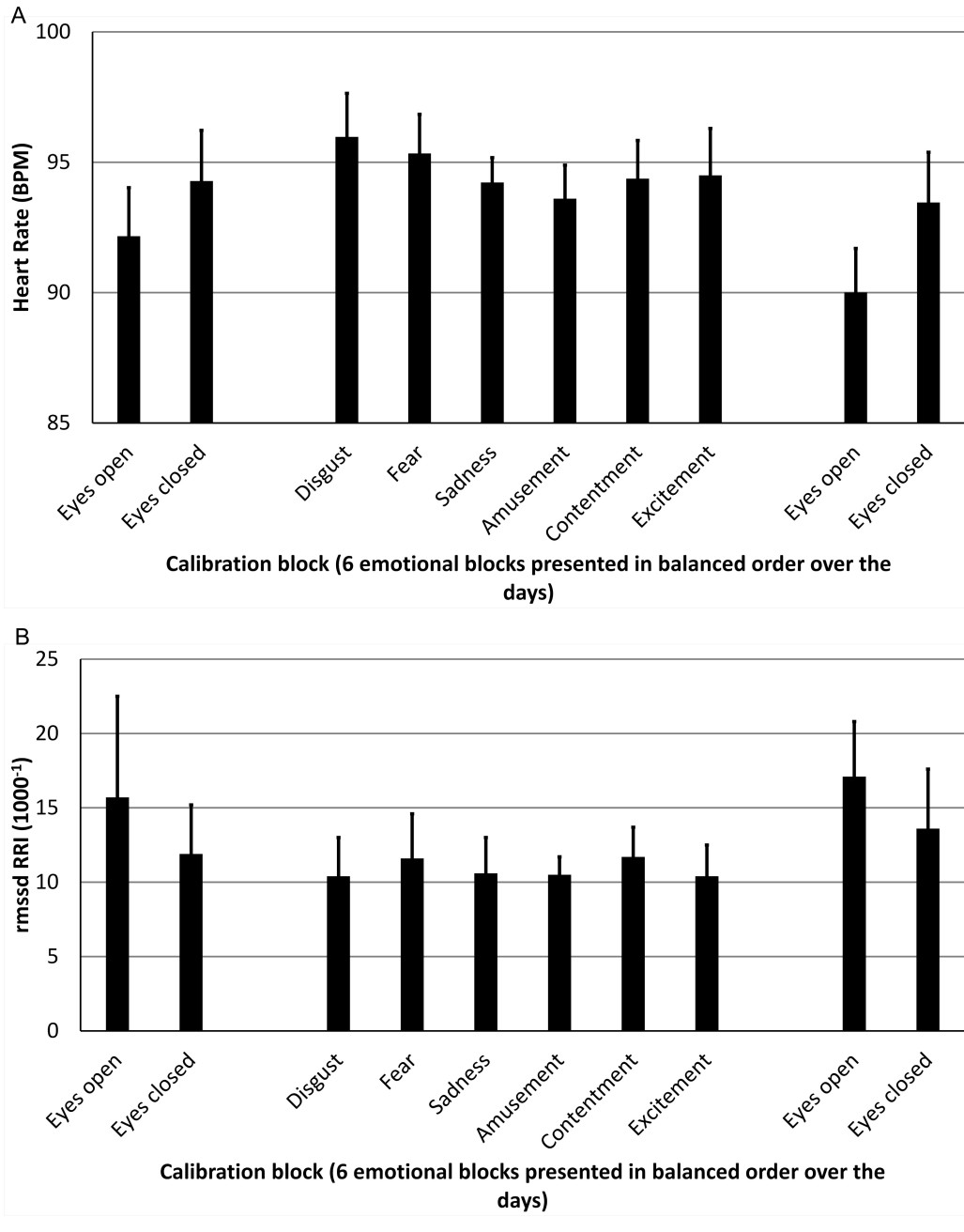

**Figure 3   Results of the physiological measures Heart Rate (A) and rmssd RRI (B) as function of the different calibration blocks.** Eyes open and Eyes closed blocks were measured before and after the emotional blocks. The order of the emotional blocks was balanced over days. Error bars denote the standard error of the mean.

variable showed significant differences for the heart rate variability measures only (all *F* values > 5.83, all *p* values < .02). Figure 3 gives the HR and rmssd RRI as function of calibration block.

Figure 4 summarizes the power distribution for the different frequency bands averaged over the rest epochs (Fig. 4A) and the writing epochs (Fig. 4B). When the EEG features
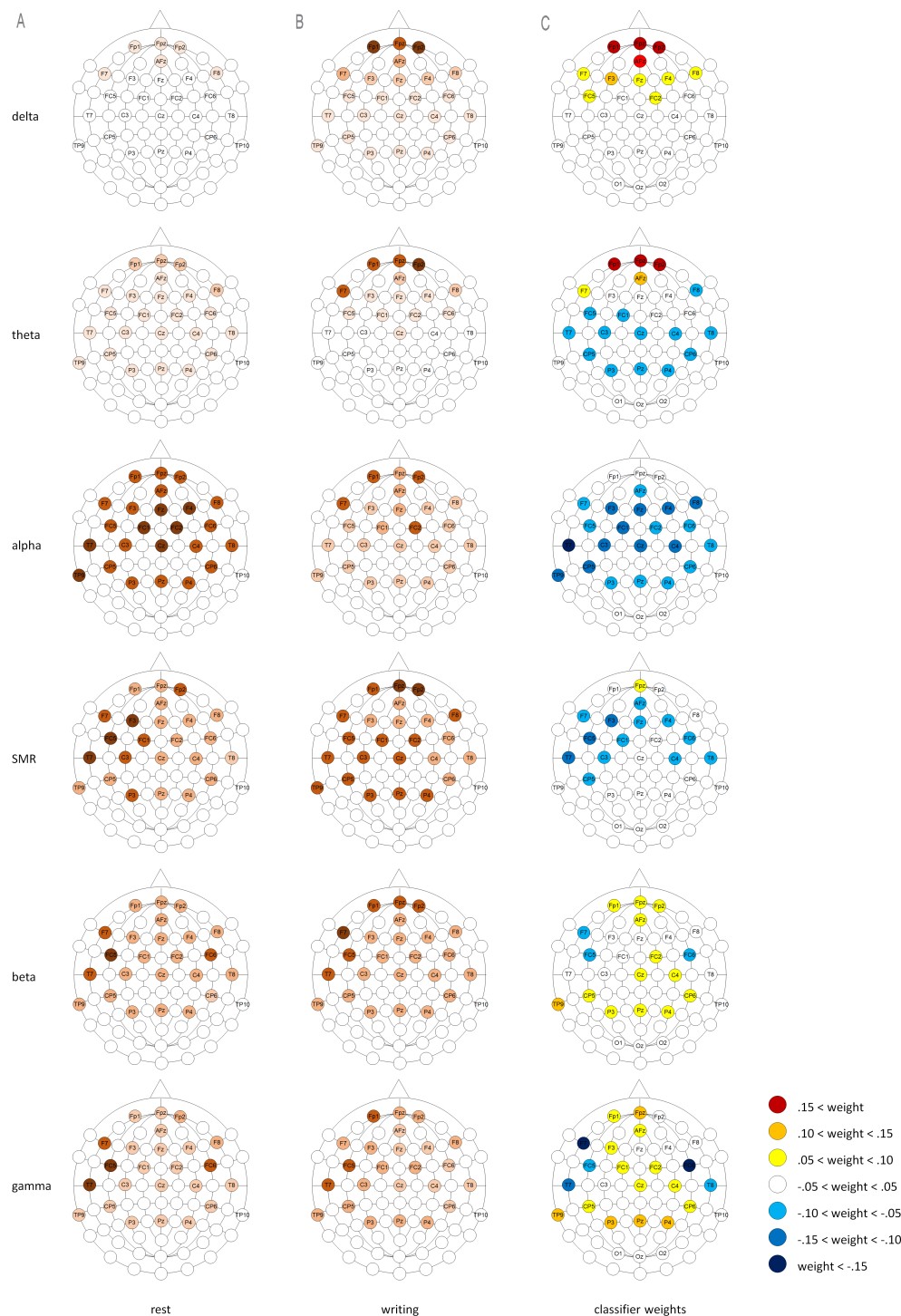

**Figure 4** Column (A) shows the power in the different frequency bands in the rest blocks (eyes open and eyes closed combined) and column (B) in the writing blocks. Column (C) gives the weights of the features in the classification model.

were used as input, the average model performance (over all days) was 92% with a hit-rate of 86% and a False-Alarm-rate (FA-rate) of 4%, resulting in an equal cases performance of 91% ($p < .01$). The model weights are depicted in Fig. 4C. Inspecting Fig. 4 shows that there are three major differences between rest and writing reflected in the model weights. The writing blocks show: (1) increased frontal power in the delta and theta bands, (2) wide spread suppression of alpha, and (3) central increase in beta and gamma activity. The first effect is most likely caused by eye movements. The second effect relates to the suppression of the brain's idle state during rest. The increase in gamma activity may be related to creative processes as described in 'The Creative Brain', but gamma components are also susceptible to muscle artifacts caused by e.g., jaw clenching and forehead movements. If we only use the alpha and gamma features in the classifier, equal class performance is 88%, indicating that a reliable difference can be obtained without using features that may be contaminated with eye movements.

When all features (peripheral physiology and EEG) were used as input the average model performance was also 92%, with a hit-rate of 89% and a False-Alarm-rate (FA-rate) of 6%, resulting in an equal cases performance of 92% ($p < .01$), a non-significant improvement relative to using only EEG features, indicating that the added value of incorporating features other than EEG ones is small in this case.

A closer inspection of the feature weights in the classification model showed that the highest weights are attributed to the delta (0–4 Hz) and theta (4–8) bands in channels Fp1, Fpz, Fp2 (i.e., frontal channels). The equal class performance of a classification model using only these six features is 0.84 (compared to 0.92 for a model using all features). Slow (0–4 Hz) frequency bands of EEG may pick up eye movements and should be evaluated with caution (please note that eye movements were not removed from the EEG data). Indirect measurement of eye movements in the EEG signal masks the information in the primary EEG. Even in case it is a reliable classifier for the current experimental setup, we consider it an artifact.

### Classification of valence and arousal in the calibration blocks

Model performance was determined for classifying low vs. high valence and arousal using 10-fold cross validation using a range of parameters:

- Features from EEG, physiology or both,
- Binary classification for predicting outcomes higher than the median value or using only the extreme values, i.e., lower than the 0.33-quantile or higher than the 0.66-quantile,
- Using raw or normalized features, in which case the features were normalized by dividing by the average feature value for the Eyes-Open conditions (for that day),
- Using SVM or elastic net classifier with logistic regression.

In none of the cases did we find classification performance deviating significantly from chance performance. Since classification performance using the whole set of EEG data did not result in above-chance performance, we did not continue using specific subsets only, e.g., to look at the power in specific EEG frequency bands like alpha (*Bahramisharif et al., 2010*; *Klimesch, Sauseng & Hanslmayr, 2007*), at the relative power in different EEG bands (*Herbert, Junghofer & Kissler, 2008*) and at asymmetrical alpha activity in the prefrontal

cortex. Individual ANOVAs on the physiological measures confirmed these observations: all $F$-values $< 0.63$ and all $p$ values $> .67$; see also Fig. 3. Because building a reliable valence and arousal classification algorithm using the calibration data turned out to be impossible, we could not further classify the novel writing data.

### Facial expression in the calibration blocks

We used the FaceReader® output directly in the analysis and found no significant differences between the different emotional paragraphs. Generally, the facial expression of the writer was classified as neutral (about 30%), sad (about 25%) or angry (about 20%). The remaining 25% was dispersed over happy, surprised, scared and disgusted.

### Subjective questionnaires

The DES full showed neither differences over the days (1–9) nor over sessions (start—end of day). Analysis of the feelings grid scores showed a significant effect of arousal over sessions: $F(3, 31) = 4.57$, $p < .01$. A post-hoc LSD test showed a significant difference between start of the day and the end of block 2 and end of the day. The analyses of the VAS scores showed no effect over days, but a large effect over sessions of happy: $F(3, 31) = 3.65$, $p < .03$, optimistic: $F(3, 31) = 6.28$, $p < .01$ and flow: $F(3, 31) = 6.76$, $p < .001$, and a trend for relaxed: $F(3, 31) = 2.38$, $p < .09$. The means of the significant effects over session are presented in Fig. 5. The figure shows that happy, optimism and flow are rated high at the start of the day but systematically decrease over the writing sessions with a stabilization or reversal at the end of the day. For arousal, this effect is inverted. These trends are confirmed by post-hoc LSD tests.

In the daily debriefing session at the end of the day, the writer indicated that the EEG cap was uncomfortable at the start but that he got used to wearing the cap and the other physiological sensors. He experienced the cameras as more obtrusive and disturbing than the physiological sensors. He elaborated on this in several public interviews (e.g., in The New York Times: www.nyti.ms/1dGxkFR).

## Discussion
### Set-up and user experience

The case study primarily focussed on measuring neurophysiological indices over prolonged periods outside a laboratory environment before applying the technology in a large scale experiment with the readers of the novel. Inspection of the signals revealed that, except for EEG channels O1, Oz and O2, we were able to record reliable signals in a real life situation using wearable/wireless sensor technology and that the setup was comfortable enough for the writer to work for hours a day wearing the sensors. The noise in the occipital channels may be caused by (neck) muscle activity related to mouse and keyboard actions. The ICA analysis indicated that potential artifacts were non-stationary (i.e., changed over time), an effect similar to what we find with readers (*Brouwer et al., 2015*). Non-stationarities may be more common in real-world, multitasking environments and hamper identification and removal of artifacts (*Van Erp, Lotte & Tangermann, 2012*). This increases the relevance of including EMG and EOG sensors to the sensor suite. Data analysis may also benefit from a higher electrode density allowing to apply more advanced techniques for artifact removal

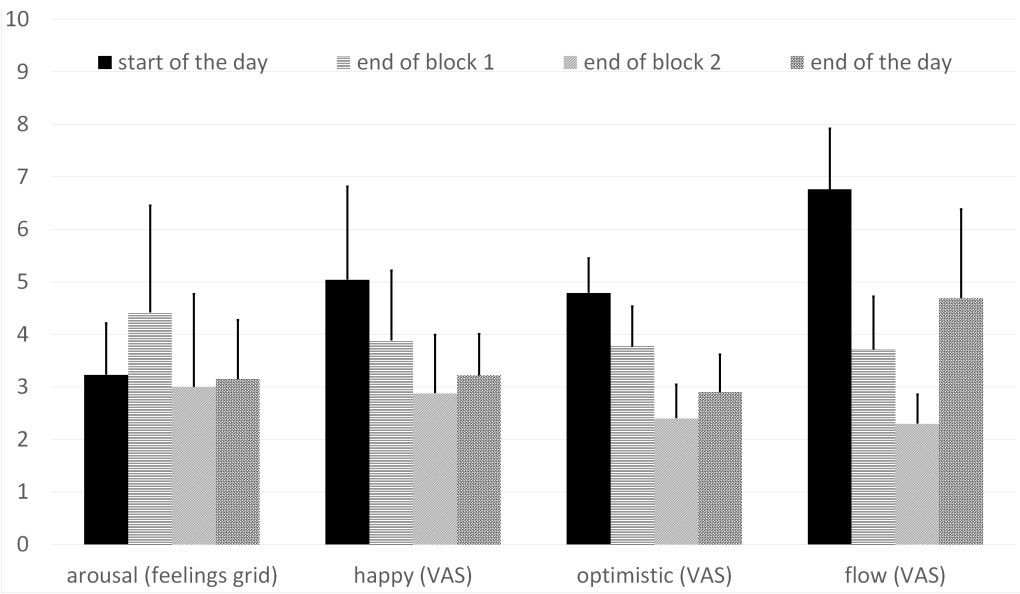

**Figure 5** Significant changes in subjective ratings over the course of a writing day.

and EEG analysis. Recent electrode developments may enable this without reducing usability and comfort over prolonged periods of use.

Although measuring physiology outside a well-controlled laboratory environment is challenging, the data show reliable differences between resting state and writing, which indicates a sufficient signal-to-noise ratio in the data. It could still be the case that this "writing detector" is triggered by artifacts like eye movements or muscle activity that comes with typing (however, the EEG channels most prone to these muscle artifacts (O1, Oz and O2) were removed from the dataset). If we look at the weight of the different features in the classification algorithm, we see that most weight is attributed to the delta (0–4 Hz) and theta (4–8) bands in the frontal channels (Fp1, Fpz, Fp2). Low frequency bands should be evaluated with caution as they may reflects eye movements rather than signals in the primary EEG. This is especially relevant for the current dataset since (eye movement) artifacts were non-stationary and could not be reliably identified and removed. However, the classifier is also based on suppression of alpha and increased central gamma activity during writing. This matches with the expected pattern for creative writing although one should note that gamma can also be affected by e.g., jaw clenching artifacts. In addition, the current differences in physiology like increased heartrate and decreased heartrate variability (see Fig. 3) do fit with an interpretation of low (rest) and high cognitive activity (writing) and not just with simple muscle activity. To exclude the aforementioned artifacts, a comparison should be made between writing about an emotion, writing down mundane instructions, and for instance copying text or making random typing movements.

### Neurophysiology of emotional writing

Since there are no data available about neurophysiological correlates of emotional writing, we based our expectations on research into physiological responses based on presenting

many repetitions of, for instance, emotional pictures or sounds. In this domain, recent experiments show that changes in emotional state can also be reliably identified with a restricted number of repetitions or even single trial (especially for longer epochs like in our study). Therefore, we expected to be able to see changes in physiological state as a function of emotional content, despite the limited number of repetitions. However, we were not able to link specific neurophysiological indices to the emotional content of the writing. We have three possible explanations: (1) the quantity and/or quality of the data was not sufficient, (2) writing is a cognitive rather than an emotional task for this particular author, and (3) the task involved a multitude of emotional, creative and cognitive processes concealing the single-task indices found in single-task laboratory experiments. The first explanation pleads for expanding the data set using more authors and possibly more sessions than we were currently able to gather. Nevertheless, we should keep in mind that the current data was sufficiently reliable to classify rest from writing with 92% accuracy, and the employed classifications methods are sensitive enough to be used on smaller datasets. This forces us to look into alternative explanations as well before upscaling. One such explanation is that for this particular writer, the writing process itself may predominantly be a cognitive task and unrelated to the emotional content, i.e., the writer does not experience a particular emotion himself when writing about it. The neurophysiological pattern found in writing compared to rest and the facial expression (often classified as neutral) fit with the signature of a cognitive task. Based on the vast production of the writer and as confirmed in later discussion with him, this is a viable option. In hindsight, the time pressure (2 min per item), the strict instruction (write about this particular emotion fitting with this particular picture), the time of day (always in the morning before the writing block started), and the presence of the experimenter may all have triggered cognitive controlled creativity rather than emotional or spontaneous creativity. The third factor that may have played a role in the current results is the task setting that may have resulted in multiple processes (including but not limited to emotional, associative, creative, linguistic and motor planning processes). The resulting brain activity patterns may not be comparable to those for passive viewing of emotional pictures in a laboratory environment.

### Subjective ratings

The ratings of arousal, happy, optimistic and flow seem to show the same pattern. At the start of the day, the writer is in a 'relaxed, good mood' but his mood seems to dwindle during the writing with increasing arousal. At the end of the day, after the last writing session, this pattern stabilizes or is reversed. This profile in part reflects the circadian modulation of mood and related aspects.

## THE EBOOK OF THE FUTURE

One may ask if uncovering brain states associated with art will de-mythologize the process: will art lose its meaning, beauty or purity when reduced to activity of groups of neurons? Will we eventually reveal the mechanisms of art and thus render it mechanical? Will scientists be able to develop a drug that makes everyone a best-selling author? Will this knowledge increase the 'creativity rat race' for artistic and creative success as cognitive

enhancers may do in the 'cognitive rat race' in the academic world (*Repantis et al., 2010*)? We think not—but raising and discussing these questions is of utmost importance for the field (*Van Erp, Lotte & Tangermann, 2012*). A more interesting debate is whether creative writing is a skill one can develop like skilled behavior in sports and music, or possibly even non-creative writing like scientists and journalists do on a daily basis. Creative skills are important outside the arts and the creative industry and their importance is widely acknowledged in an innovative and knowledge-based economy. We would like to expand our research into (spontaneous) creativity to answer important questions and develop appropriate tests and tools to measure spontaneous creativity (which may require 24 h measurements).

Current ebooks have the ability to track reader behavior and ebook retailers are actively gathering (anonymous) data of their readers on parameters such as the books the reader has finished (or not), how fast, where reading was discontinued and for how long and which words were looked up in a connected dictionary (*Flood, 2014*). None of this information is directly used for the benefit of the reader but serves manufacturers and publishers only. The basis for our approach is to measure the readers' state and behavior to make them the primary beneficiaries, for instance through enhancing the reader experience. There are many approaches foreseeable. A relatively simple one that is not interactive yet is to use the emotional response to give better informed advice on other books the reader may enjoy. In a similar way, readers may want to share their emotional profile, for instance by posting it on social media or through new communities of people with similar frames of mind around a specific book. Real interactivity may also come in many forms. For instance, the emotional response may be used to add music or other multisensory stimuli to further intensify the experience or ultimately change the storyline or the flow of the book. This may lead to new media products that are somewhere in between literature, movies and games.

## ACKNOWLEDGEMENTS

We kindly acknowledge the great help of Christian Vermorken and Marc Grootjen from Eaglescience, Andrew Spink from Noldus IT, and Leo Hoogendoorn from TMSi for providing hardware and software components and helping us with the measurements and analyses. We sincerely thank Arnon Grunberg and his publishers Elik Lettinga and Paulien Loerts from Nijgh & Van Ditmar for sharing their time and their creative minds.

### Funding
The authors received no funding for this work.

### Competing Interests
The authors declare there are no competing interests.

## Author Contributions

- Jan B.F. van Erp conceived and designed the experiments, performed the experiments, contributed reagents/materials/analysis tools, wrote the paper, prepared figures and/or tables, reviewed drafts of the paper.
- Maarten A. Hogervorst analyzed the data, contributed reagents/materials/analysis tools, wrote the paper, prepared figures and/or tables, performed the computation work, reviewed drafts of the paper.
- Ysbrand D. van der Werf conceived and designed the experiments, performed the experiments, analyzed the data, contributed reagents/materials/analysis tools, wrote the paper, prepared figures and/or tables, reviewed drafts of the paper.

## Ethics

The following information was supplied relating to ethical approvals (i.e., approving body and any reference numbers):

The Institutional Review Board of TNO Human Factors (TCPE Soesterberg, The Netherlands) approved this study.

## Data Availability

This article describes the data of a single participant who is mentioned by name. To protect his privacy (including full address), the raw data will only be available to those signing a non-disclosure agreement with the first author's institution (TNO). Please contact the corresponding author for more information.

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
