# Peer review of "Toward physiological indices of emotional state driving future ebook interactivity"

_PeerJ Computer Science, doi:10.7717/peerj-cs.60_

## Round 0.1 · original submission · Minor Revisions

I agree with the reviewers that this is an interesting and well constructed study. I suggest that you pay particular attention to addressing the following reviewer comments:

• Reviewer 1 points out that the title refers to the final goal of the research rather than the results of this particular study.

• Both Reviewers 1 and 2 suggest clarifications and additions to improve the presentation and explanation of results of the study.

• Reviewer 1 suggests a discussion of how this study relates to the work presented in Onton & Makeig 2009.

Reviewer 1 suggests that the literature review be greatly reduced and perhaps be reword as a separate review article. Reviewer 2 and I do not feel strongly about the length of the review section, so I will leave it to your decision as to whether you wish to retain the literature review in this present article or reduce it.

·

Basic reporting

In this study the authors attempt to determine the affective or emotional state of a well know writer, engaged in original composition, from a range of physiological and neurophysiological signals. The authors attempt to induce and classify emotional states of the writer in response to an IAPS picture set. While successful at separating calibration from writing sessions, the authors were not able to isolate individual emotional states. The research presented in this manuscript is the beginning of an ambitions study to identify/classify the emotional state of readers in responses a written (emotional) narrative. The ultimate goal being an ebook-like system that can adapted to the mental state of the individual reader.

Experimental design

The work described in this manuscript was conducted with an appropriate amount of scientific rigor, given the real-world task and environment. As such, it will contribute to the growing field of applied neuroscience and real-world neuroimaging. However, there are several revisions that would significantly improve this manuscript (see below).

Validity of the findings

The title and abstract are misleading and imply that this study develops/uses an interactive ebook system – this text must be revised. The final use of this technology can be discussed in the intro, or more appropriately, the discussion section. Related to this comment is the structure of the manuscript, which at first reads like a review/theory article. In the introduction alone there are sections that cover ebooks, art and neuroscience, psychology and neurophysiology of emotions (valence and arousal), as well as cognitive processes that underlie creativity. This is too verbose an introduction for a study that only resulted in a differentiation between calibration and writing sessions. While optional, I suggest the authors split this manuscript into a review article and a shorter case study.

The other major concern is related to the neurophysiological analysis described in the manuscript. While the ultimate goal of the study was to classify emotions, the manuscript would be improved by adding a figure that outlines the EEG components that were most discriminative between calibration and writing sessions. What were the most discriminative features/electrodes? This is also related to my concern with the artifact processing steps described in the manuscript. While individual electrodes with high variance (e.g. Oz) were excluded, there appears to be no attempt at removing 1) EoG components and 2) epochs of EMG or other noise artifacts. How can the authors be sure that the difference between calibration and writing sessions was not due to eye or muscle movements (e.g. picture scanning or jaw clenching)? Eye movement metrics especially have been shown to differentiate between even relatively similar tasks, with blinks and saccades often reflected the Alpha spectrum of frontal channels. The manuscript would be significantly improved if these concerns were addressed with the addition of EEG spectral and/or summary figure(s).

Given the results, it would seem prudent for the next study to have a control component/condition that includes non-emotional writing or non-cognitive writing (e.g. copying) to see if these sessions can be differentiated from the emotional ones.

The authors should address the relationship of the present study to Onton & Makeig 2009 (“High-frequency Broadband Modulations of Electroencephalographic Spectra”). Would the author’s classification approach have been more successful if they had used methods similar to Onton & Makeig

The subjective ratings (figure 4) and the participant’s commentary about the comfort and obtrusiveness of the monitoring systems was very informative and relevant for future research in this area.

Additional comments

Minor Points:

The majority of the (vertical) figure axes are unlabeled or without units. Please include labels and units in future revisions.

Typos and Grammar:

Line 36: change to "Sales of ebooks are..."
Line 175: add commas "...through, for instance, skin..."
Lines 173 through 179: Sentence is unwieldy. Please break into parts or reduce the examples.
Line 188: Begin sentence with "An important step..."
Line 189: either remove the word "potential" or the word "added"
Line 190: add a comma "...is new, and entertainment..."
Line 192: change "artefacts" to "artifacts"
Line 196: remove the word "use"
Line 216: change to "This is a powerful mechanism..."
Line 256: change to "Arnon Grunberg was..."
Line 483: "scientist" should be plural

·

Basic reporting

van Erp et al. suggest an interactive form of ebooks. The story would take into account physiological signals of the reader and thereby create an interactive format. Tests of the concept on the author during writing were partly successful, differentiating writing phases from non writing phases. But it was not possible yet to differentiate parts of the story with different emotional content. The manuscript presents an intriguing idea and is well written.

Introduction
line 51 - include an up to date authoritative review, e.g. Dehaene et al. Nature Review Neuroscience 2015
line 74 - recording EEG and further physiological measures of several hundreds of subjects is a lot of work. Make such a promise only if you really follow up on it.
line 100 - this section is devoid of references. Add at least one, e.g. Dalgleish Nature Review Neuroscience 2004 or Tovote et al 2015
line 129 - a side remark: contentment was not listed as a primary emotion.
line 171 - Arousal and emotions also have a significant influence on eye movements. This in turn is important for EEG measurements e.g. Kaspar and König 2012 Front Hum Neurosci

Experimental design

Methods
line 269 - the setup is nice, I like it.
line 298 - You should separate eye movement induced artefacts in the EEG recordings by ICA based methods as in Plöchl et al 2012 Front Hum Neurosci. In principle you can live with indirect measurement of eye movements in the EEG signal, but then you lose most of the information in the primary EEG. In the present case the number of electrodes is moderate and it has to be tested whether enough channels remain to make statements on e.g. DLPFC. At the very least, this has to be discussed. It might contribute to the high performance of EEG features in line 377 or low performance in line 386.

Validity of the findings

line 351 - The statistical methods are valid

Results
ok

Additional comments

Discussion
line 429 - The EEG recordings might be reliable, but wether it is really EEG is a different question. Same issue as above, automated cleaning/separating algorithms are available.
line 462 - “i.e. the writer does not experience a particular emotion himself when writing about it. “ is this so?
The discussion could address methodological aspects. For example increasing the number of EEG channels could be really helpful.

signed Peter König

---

## Round 0.2 · Minor Revisions

While this revised draft has gone some way towards addressing the reviewers' initial recommendations, there are still issues with the literature review; both reviewers point to further refinements that are necessary. Further, some of the images in the figures are not of the best quality.

·

Basic reporting

In this study the authors attempt to determine the affective or emotional state of a well know writer, engaged in original composition, from a range of physiological and neurophysiological signals. The authors attempt to induce and classify emotional states of the writer in response to an IAPS picture set. While successful at separating calibration from writing sessions, the authors were not able to isolate individual emotional states. The research presented in this (revised) manuscript is the beginning of an ambitions study to identify/classify the emotional state of readers in responses a written (emotional) narrative. The ultimate goal being an ebook-like system that can adapted to the mental state of the individual reader.

The manuscript meets all the PeerJ basic reporting standards.

Experimental design

The work described in this manuscript was conducted with an appropriate amount of scientific rigor, given the real-world task and environment. As such, it will contribute to the growing field of applied neuroscience and real-world neuroimaging.

Validity of the findings

Minor Concerns:

Some aspects of the introduction present neuroscientific theories or frameworks (e.g. the emotional brain) as more established than they currently are. Many of the components of the introduction (emotion, beauty, creativity) describe research at the forefront of cognitive neuroscience and are, at present, vigorously debated.

The authors are appropriately guarded and circumspect in their interpretation of the frontal delta and theta components. Indeed, these are likely to be eye movement related features. However, the gamma components are also likely to be (at least partially) driven by EMG. Even though the occipital channels were removed to mitigate neck muscle artifacts, jaw clenching and forehead movements can also be reflected in the gamma bands across the scalp.

The embedded image quality was very low making some of the figures very hard to interpret (especially new figure 4). Hopefully, this will be resolved during the publication process.

Additional comments

The revisions in response to the reviewers concerns have significantly improved the manuscript. The (substantial) introduction now provides an appropriate background and literature review for this very ambitious study. The abstract, introduction, results and discussion now flow and provide a more coherent narrative. The expanded results likewise provide more insight into the EEG classification component. My previous concerns have been address and I now support publication of this manuscript.

·

Basic reporting

In the first round i gave a rather positive evaluation. With the revision, however, I'm not impressed. The authors were very reluctant to include the suggstions and the manuscript has not been imporved.

The relation to the existing literature is not adequat. Yet, the authors step up their claim by including the claim that "We describe the theoretical foundation of the emotional and creative brain and review the neurophysiological indices that can be used to drive future ebook interactivity in a real life situation." If this has aspects of a review, then it is mandatory that the current state of the art is referenced and the claims made adapted to what is delivered in the present manuscript.

The nature of the signal measured is a central question. In EEG measurements many artifactual sources are known. Eye movements are a prominent example. To state (line 406) "Initialy, the EEG data of the remaining channels were used in an Independent Component Analysis (ICA) to identify and remove potential artifacts. However, the ICA revealed that potential artifacts were non-stationary (i.e. changing over time) and therefore difficult to identify and thus no more data were removed. " or in short, there are so many artifacts, we leave them in, is not good enough. I'd expect cleaning the EEG data of artifacts, or in the very least a thorough discussion of the issue with appropriate references to state of the art techniques.

Otherwise it is not clear whether the recordings refer to brain activity (as implied by EEG and BCI) or to eye movements or muscle activity.

Experimental design

...

Validity of the findings

As long as the issues above are not adequatly addressed, i.e. wether brain activity has been recorded, the validity of the report is questionable.

Additional comments

...

---

## Round 0.3 · accepted · Accept

These final revisions have addressed the remaining concerns of the reviewers. Thank you for your work in revising the article.